# Do Automatic Factuality Metrics Measure Factuality?
# A Critical Evaluation

**Sanjana Ramprasad**
Northeastern University
ramprasad.sa@northeastern.edu

**Byron C. Wallace**
Northeastern University
b.wallace@northeastern.edu

## Abstract

Modern LLMs can now produce highly readable abstractive summaries, to the point that traditional automated metrics for evaluating summary quality, such as ROUGE, have saturated. However, LLMs still sometimes introduce inaccuracies into summaries, i.e., information inconsistent with or unsupported by the corresponding source. Measuring the occurrence of these often subtle factual inconsistencies automatically has proved challenging. This in turn has motivated development of metrics intended to measure the factual consistency of generated summaries against sources. But are these approaches measuring what they purport to? Or are they mostly exploiting artifacts? In this work, we stress test a range of automatic factuality metrics, including specialized models and LLM-based prompting methods, to probe what they actually capture. Using a shallow classifier to separate "easy" examples for factual evaluation where surface features suffice from "hard" cases requiring deeper reasoning, we find that all metrics show substantial performance drops on the latter. Furthermore, some metrics are more sensitive to benign, fact-preserving edits than to factual corrections. Building on this observation, we demonstrate that most automatic factuality metrics can be gamed, i.e., their scores can be artificially inflated by appending innocuous, content-free sentences to summaries. Among the metrics tested, the LLM-based ChatGPT-DA approach is the most robust and reliable. However, this comes with a notable caveat: Prompting LLMs to assess factuality may overly rely on their parametric knowledge rather than the provided reference when making judgments. Taken together, our findings call into question the reliability of current factuality metrics and prompt a broader reflection on what these metrics are truly measuring. We conclude with concrete recommendations for improving both benchmark design and metric robustness, particularly in light of their vulnerability to superficial manipulations.

## 1 Introduction

LLMs are strong abstractive summarizers [Goyal et al., 2022, Zhang et al., 2024], but they are not infallible. Even the largest, most capable models sometimes introduce subtle "hallucinations" (or "confabulations") into summaries that are unsupported by or in contradiction to the corresponding input document [Zhang et al., 2024, Tang et al., 2024b, Ramprasad et al., 2024b]. Such behavior is especially problematic in domains such as medicine or law, where inaccurate information could translate into meaningfully negative consequences for individuals.

However, manually evaluating model outputs' factual consistency with respect to references is expensive, time-consuming, and impractical to scale. This has motivated development of automated methods that score generated summaries for consistency with respect to reference documents. Such metrics have been operationalized using a range of techniques, including entailment (SummaC; Laban et al. [2022]), QA models (QuestEval; Scialom et al. [2021]), specialized models explicitly trained to score source-summary pairs (UniEval; Zhong et al. [2022], Alignscore; Zha et al. [2023],

39th Conference on Neural Information Processing Systems (NeurIPS 2025).

MiniCheck; Tang et al. [2024a]), and more recently, LLM-based methods that rely on prompting LLMs (ChatGPT-DA; Wang et al. [2023a]; Luo et al. [2023]).

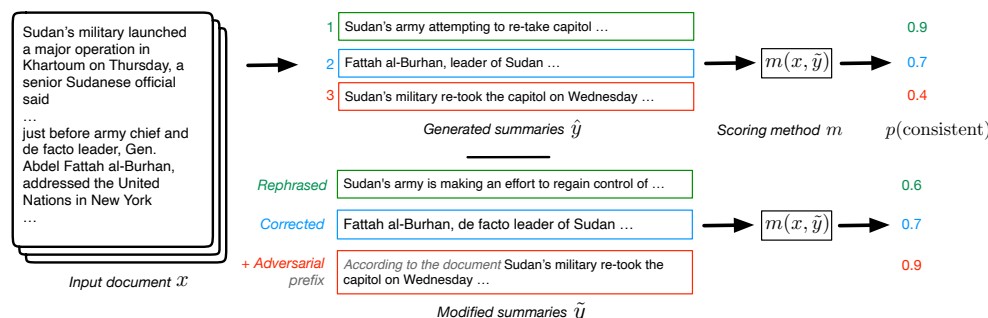

Figure 1: Many methods have been proposed to automatically evaluate the factual consistency of summaries with respect to inputs. In this work we critically evaluate such approaches, e.g., by measuring their sensitivity to various manipulations, as shown here.

Most of these metrics have been evaluated against human benchmark assessments (binary labels or Likert scores) of factual consistency [Maynez et al., 2020, Fabbri et al., 2021, Laban et al., 2022, Honovich et al., 2022, Wang et al., 2022, Gao et al., 2023, Tang et al., 2024a].

These assessments have established that automated factuality metrics correlate with human evaluations to varying degrees. But are such approaches actually attuned to the subtle question of factual consistency between inputs and outputs, or are they merely relying on shallow heuristic signals to make their judgments?

If metrics do rely on shallow heuristic patterns, this raises questions about their reliability. It also makes them vulnerable to "gaming": Summaries or claims can be crafted to exploit heuristic shortcuts and artificially inflate scores without genuinely improving factual accuracy. Alternatively, LLM-based approaches may overly rely on their internal (parametric) knowledge to assess summaries, rather than assessing output factuality with respect to the source. In this work, we stress test a diverse set of SOTA factual consistency metrics on context-claim pairs that yield continuous factual consistency scores. Our analyses offer the following empirical results as our main contributions.

**Metrics struggle when factuality requires reasoning beyond shallow cues.** In Section 3 we train a shallow MLP on surface-level heuristics and use its prediction confidence to categorize summaries as easy, moderate, or hard to assess for factuality. We find that all automated metrics exhibit substantial performance drops from easy to hard examples, suggesting they may be relying on superficial cues rather than nuanced reasoning.

**Factuality metrics are somewhat responsive to factual corrections, but also often sensitive to irrelevant (benign) modifications to summaries.** In Section 4.1, we evaluate whether factuality metrics distinguish genuine factual corrections from inconsequential edits using a dataset of annotated summary pairs [Krishna et al., 2024] which comprise an inconsistent summary and a minimally edited, faithful revision. Ideally, metrics should assign higher scores to the corrected versions while remaining stable under benign, fact-preserving edits (e.g., paraphrasing). We find that while both specialized and prompt-based metrics respond to factual corrections, many—especially NLI-based SummaC and some specialized models (UniEval and Alignscore)—are overly sensitive to benign edits. In fact, some metrics exhibit greater score shifts from superficial changes than from actual factual improvements. By contrast, ChatGPT-DA consistently ranks faithful revisions higher while remaining robust to benign perturbations, suggesting that it may be a more reliable choice.

**(Some) factual consistency metrics are gameable.** If factuality metrics rely at least partially on superficial cues, this suggests that we should be able to "game" them by inserting such cues to inflate scores assigned to model outputs. And indeed in Section 5 we find that inserting superfluous, innocuous phrases—either alone or appended to claims——can significantly inflate factuality scores. For NLI-based and specialized metrics, these artificial boosts often exceed the score gains achieved by genuinely more factual models (see Figure 6). ChatGPT-DA, however, shows minimal sensitivity

to such manipulations, again suggesting that prompting LLMs for factuality assessment may offer more robust and reliable evaluations. However, this comes with a caveat.

**GPT sometimes over relies on internal knowledge when evaluating summaries—even when a reference source is provided.** ChatGPT-DA may default to its internal (parametric) knowledge rather than relying on the provided source when assessing factuality consistency (see Section 4.2). This tendency appears especially pronounced when the reference document contradicts the model's parametric knowledge, indicating that its reliability may be compromised in cases involving myths, rare or updated facts, or information that conflicts with what the model "believes". This provides a caveat to our other findings that suggest LLM based assessments may be more reliable than bespoke models: LLM-based evaluations may reflect what the (evaluator) model "knows" rather than what the source says, limiting reliability in grounded factuality assessments.

**Practical recommendations.** Our analyses reveal that while specialized metrics perform well overall, they are vulnerable to benign edits and adversarial manipulation, which may limit their reliability. ChatGPT-DA offers a more robust alternative, but may emphasize consistency with its parametric knowledge rather than input sources, raising concerns about its grounding behavior. We therefore recommend caution when applying such metrics in domains involving myths, misinformation, or uncommon facts. Finally, we highlight the need for benchmarks that capture hallucination severity and for methods that incorporate saliency-aware supervision to improve metric reliability (Section 7)

## 2 Experimental Setup

### 2.1 Benchmark Datasets

Automatic factuality metrics for fact verification have been evaluated using various benchmarks, predominantly based on news sources [Hermann et al., 2015, Narayan et al., 2018]. The `AggreFact` dataset [Tang et al., 2022] consolidates several benchmarks on fine-tuned model-generated summaries from such sources [Maynez et al., 2020, Wang et al., 2020, Pagnoni et al., 2021, Fabbri et al., 2021, Honovich et al., 2022, Laban et al., 2022]. Similarly, datasets like `FacEval` [Wang et al., 2022] and `ReferenceMatters` [Gao et al., 2023] benchmark dialogue summarization of fine-tuned models on sources from `SAMSum` [Gliwa et al., 2019a] and `DialogSum` [Chen et al., 2021b].

These datasets primarily focus on summaries from fine-tuned models predating modern LLMs, which excel in zero-shot summarization [Goyal et al., 2022]. Consequently, factuality metrics evaluated on these benchmarks may not generalize to SOTA LLM outputs, which exhibit different error patterns [Tang et al., 2022]. To address this, Tang et al. [2024a] introduced `LLM-AggreFact`, a fact-checking dataset from recent LLMs across diverse domains and dataset types. Similarly, Krishna et al. [2024] released the `GenAudit` dataset with factuality annotations for LLM summaries in news, Reddit, and clinical settings, while Ramprasad et al. [2024a] focus on LLM generated dialogue summaries in the `LLM-dialogue` dataset.

For our analysis we use all of the above benchmarks to capture a wide range of error types. For fine-tuned model summaries, we use `AggreFact` for news and `FacEval` for dialogues. For LLM-generated summaries, we rely on `LLM-AggreFact`, `GenAudit`, and `LLM-dialogue`. We note that each benchmark consolidates multiple datasets and to ensure clean separation of distributions, we avoid overlapping *datasets* between our test and development splits. Since `AggreFact` primarily includes older fine-tuned models, we incorporate its dev set into our development data and reserve newer benchmarks for evaluation. Specifically, our dev set includes summaries from the `AggreFact` dev split, as well as XSUM and CNNDM examples from `Genaudit`, ensuring no overlap with test data. All remaining datasets are evaluated using their respective test splits. We provide a detailed breakdown of our dev and test splits in Appendix A.

### 2.2 Automatic Factuality Metrics for Summarization

We group SOTA factuality metrics into four broad methodological categories. This includes metrics based on: Question Answering (QA), Natural Language Inference (NLI), fine-tuned (specialized) models and LLM prompting methods; see Table 1.

For QA-based metrics we use QuestEval [Scialom et al., 2021]. As an NLI-based metric we use SummaC-Conv [Laban et al., 2022]. We use three specialized models for evaluation: UniEval,

| Metric | Category |
| --- | --- |
| QuestEval [Scialom et al., 2021] | QA |
| SummaC-Conv [Laban et al., 2022] | NLI |
| UniEval [Zhong et al., 2022] | specialized model |
| AlignScore [Zha et al., 2023] | specialized model |
| MiniCheck [Tang et al., 2024a] | specialized model |
| ChatGPT-DA [Wang et al., 2023a] | Prompt / LLM |

Table 1: Metrics categorized by approach and analyzed

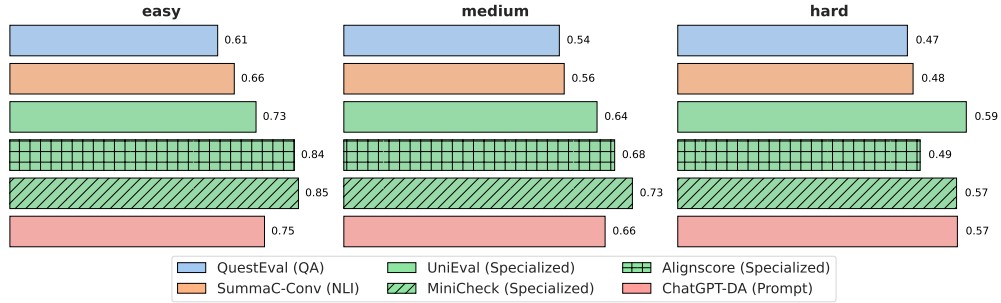

Figure 2: Summaries are categorized as easy, medium, or hard based on prediction accuracy and confidence from a shallow MLP. While specialized metrics perform best on easy examples, their performance declines on hard cases. UniEval, MiniCheck and ChatGPT-DA show greater robustness in more challenging settings

AlignScore and MiniCheck. UniEval Zhong et al. [2022] reframes NLG evaluation as a Boolean QA task and uses T5 [Raffel et al., 2020] to score different dimensions. AlignScore [Zha et al., 2023] evaluates summaries by combining an alignment function—a RoBERTa model [Liu, 2019] fine-tuned on diverse tasks—with a splitting and aggregation strategy. MiniCheck[Tang et al., 2024a] uses a Flan-T5 model [Chung et al., 2022] fine-tuned on a synthetic dataset created by the authors.

We also use GPT-4o-mini to score the factual consistency of summaries based on a direct assessment (DA) prompt template from Wang et al. [2023a].

## 3   Do metrics infer "Factuality" from Superficial Features?

The degree to which a claim is faithful to the source text is a subtle question that demands understanding the content in both texts to determine if they are consistent. We would therefore expect that metrics capable of measuring factual consistency in general would need to capitalize on information beyond the superficial features of source and claim texts just discussed.

To investigate the extent to which shallow features explain metric behavior, we train an MLP classifier to predict binary human factuality labels on a development set using only surface-level features.[1] We then apply the trained model to an evaluation set and categorize summaries into three difficulty levels—*easy*, *medium*, and *hard*—based on prediction accuracy and confidence. Confidence is measured as the absolute deviation of the predicted probability from 0.5: Lower values indicate greater uncertainty. We classify a summary as *easy* if the prediction is correct with high confidence (top 80% of confidence scores), and *medium* if correct with lower confidence. Incorrect predictions are designated *medium* if confidence is low (bottom 20%) and *hard* otherwise. The idea here is that some examples can be readily classified as "factual" (or not) using shallow features like word overlap; these are "easy" examples. By contrast, "medium" and "hard" categories capture examples that are difficult to classify with respect to factuality using these shallow features.

Figure 2 reports the AUC for metrics across summary difficulty levels. All metrics perform best on *easy* summaries, where shallow features reliably predict human judgments. Performance declines on *medium* and *hard* examples, with older metrics like QuestEval and SummaC-Conv showing sharp drops, suggesting reliance on superficial cues. Specialized and prompt-based metrics (e.g., UniEval,

---

[1]See Appendix B for feature details and Appendix C for MLP details.

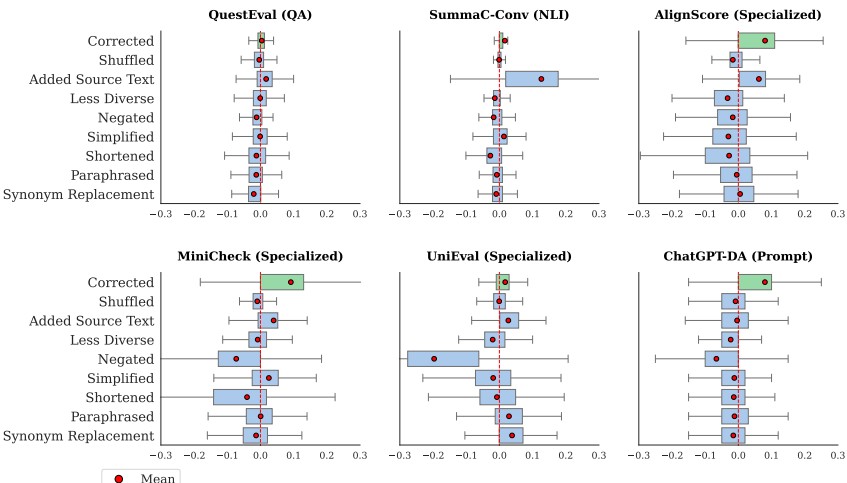

Figure 3: Score differences for each metric between the original summary and summaries edited for factual accuracy by humans (shown in green) and other benign edits (shown in blue).

AlignScore, MiniCheck, ChatGPT-DA) are more robust, maintaining higher AUC on *medium* cases. However, even these models struggle on *hard* summaries, indicating that current metrics, despite being trained for consistency, remain limited in the absence of shallow cues.

## 4  What do automatic factuality scores measure?

We next examine the reliability of factuality metrics. In Section 4.1 we evaluate their sensitivity to factual corrections versus benign (fact-preserving) edits. We find that only ChatGPT-DA consistently distinguishes between the two, showing both sensitivity and robustness. In Section 4.2, we evaluate ChatGPT-DA's reliance on source content by scoring summaries against counterfactual references that conflict with its parametric knowledge. Our results show that consistency evaluation performance degrades when presented with counterfactual input documents, indicating limited grounding in the provided sources.

### 4.1  Measuring Metric Sensitivities to Controlled Manipulation

We want to determine whether metrics respond specifically to changes in factual consistency or are influenced by superficial edits unrelated to fidelity. We focus on a subset of summaries from the `GenAudit` dataset [Krishna et al., 2024], each manually labeled as inconsistent and paired with a minimally edited, corrected version. These pairs allow us to directly assess whether metrics are sensitive to the consistency improvements that matter most.

We are also interested in score variation that owes to superfluous factors. To this end, we prompt GPT-4 to generate versions of summaries modified in targeted, fact-preserving ways intended to be independent of factual consistency. We use several prompts that request benign transformations like paraphrasing, simplification, and rewording.[2] While these edits are designed to preserve meaning, using GPT may occasionally introduce factual inconsistencies, though we believe such cases are rare.

Ideally, automatic factuality metrics should show a positive score change for corrected summaries, reflecting improved factual consistency. Conversely, generated summary rewrites—which do not alter factual content—should exhibit minimal score changes. Any significant score differences in these rewrites likely indicate that the metric is sensitive to artifacts incidental to factual consistency.

**Results.**   We present results in Figure 3. Notably, QuestEval shows no meaningful improvement in response to factual corrections, raising concerns about its ability to detect genuine consistency improvements. Additionally, most metrics, except QuestEval and ChatGPT-DA, assign dispropor-

---

[2]Prompts for benign summary edits are provided in Appendix D.

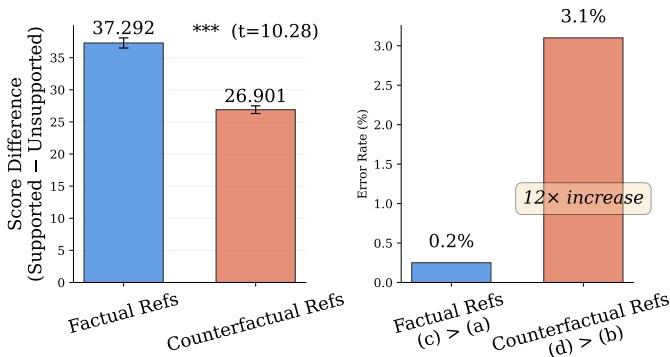

Figure 4: GPT-based consistency evaluation is influenced by parametric knowledge. Left: The score gap between supported and unsupported summaries narrows sharply when references are counterfactual but summaries are factually accurate (p < 0.001). Right: The rate of cases where unsupported summaries are scored higher than supported ones rises from 0.2% to 3.1% when references contradict GPT's world knowledge while summaries remain factually correct.

tionately high scores when random, yet consistent, source sentences are appended to inconsistent summaries.[3] In many cases, these spurious additions yield score gains comparable to, or greater than, those resulting from actual factual corrections.

We also observe that MiniCheck and UniEval are highly sensitive to meaning-preserving negation. Simple rephrasings that retain the original meaning—for example, changing "The author mistakenly believed their ACT test was on a certain day" to "The author did not realize their ACT test was not on the day they thought", often lead to reduced scores. While ChatGPT-DA exhibits some sensitivity to negation, it remains relatively robust overall, consistently distinguishing between true factual improvements and irrelevant textual changes.

## 4.2 Reliance on Source Context Versus Parametric Knowledge in GPT-Based Scoring

Prompt-based approaches to factual consistency evaluation have recently gained traction [Luo et al., 2023, Wang et al., 2023a]. Unlike traditional metrics trained on source-summary pairs, these methods rely on prompting pre-trained models to evaluate summaries against reference articles for consistency. This raises a critical question: Are their judgments truly grounded in the provided source, or are they influenced by the model's internal (parametric) knowledge?

We investigate this by evaluating GPT's consistency judgments under scenarios where the reference text (likely) conflicts with the model's implicit world knowledge. To ensure that GPT possesses strong prior knowledge about the domain, we use Wikipedia articles, which are heavily represented in LLM pretraining corpora. Specifically, we use the ConflictBank dataset [Su et al., 2024], which comprises factual claims extracted from Wikipedia and corresponding counterfactual variants generated through targeted substitutions. Each counterfactual claim is also paired with a counterfactual reference document—a version of the original article modified to make the false claim appear consistent with the altered reference. This dataset structure permits four experimental conditions: (a) Factual references paired with supported factual summaries; (b) Counterfactual references paired with supported counterfactual summaries; (c) Factual references paired with unsupported counterfactual summaries, and; (d) Counterfactual references paired with unsupported factual summaries.

To test the discriminative ability of GPT to score for consistency, we evaluate against factual references using conditions (a) and (c) and counterfactual references using conditions (b) and (d). Ideally GPT would be equally discriminative in both cases: (a) and (b) should be deemed equally consistent, and (c) and (d) be scored as equally inconsistent. In Figure 4, left, we compare the score differences between supported and unsupported summaries in both settings, namely (a - c) and (b - d), using a

---

[3]Such a sentence will itself be "consistent", but should probably not shift the overall consistency score to be higher than the corrected summary given that this does not *correct* any existing inconsistencies, as it strictly adds content.

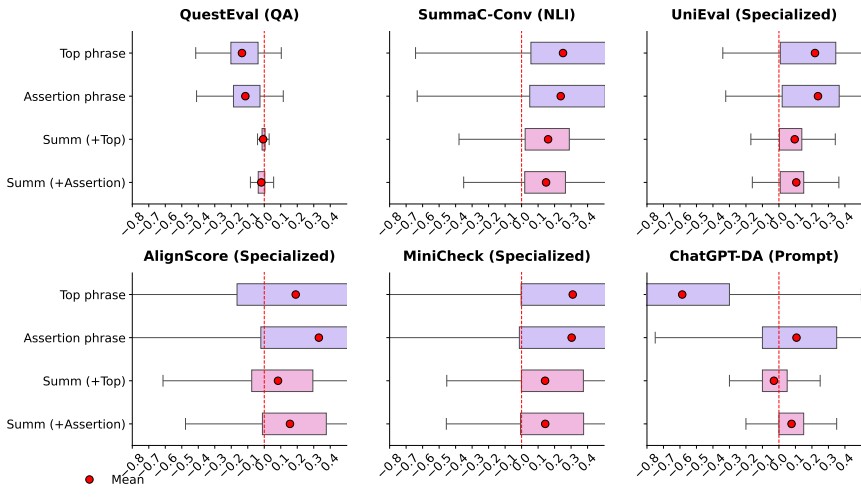

Figure 5: Pairwise score differences across summaries manipulated using four distinct strategies: Adding constant strings (**top phrase** and **assertion phrase**) and appending them to summaries (**summ + top**, and **summ + assertion**. The results reveal that NLI and bespoke model metrics are particularly vulnerable to gaming, with significant score inflation observed under these manipulations.

paired t-test on the two deltas. The test yields p < 0.001, i.e., the score gap between supported and unsupported summaries is significantly smaller when evaluated against counterfactual references compared to factual ones. This suggests that GPT struggles more to distinguish between supported and unsupported summaries when the reference contradicts its internal knowledge. In particular, when evaluating summaries against references that contradicts its internal knowledge, the model shows reduced sensitivity to whether the summary is actually supported by evidence.

We would also anticipate that the model should score supported summaries higher than unsupported summaries, i.e., b > d. However, if d > b, this might suggest that GPT's parametric knowledge is overriding the reference-based evaluation, inappropriately favoring (generally) factually accurate content which is unsupported by the input under consideration. We find that GPT incorrectly rates the unsupported (but factual) summary higher than the supported summary in 3% of cases. As a baseline, we also evaluate the proportion of times it scores unsupported outputs higher than supported factual summaries where the input is (generally) factual (c > a). Here, GPT rates unsupported summaries higher only 0.2% of the time. The increase in inconsistent summaries being rated higher $(0.2\% \rightarrow 3\%)$ suggests that GPT struggles more with consistency evaluation when references contain misinformation or contradict established facts. This suggests the model's evaluation reliability could be compromised in real-world scenarios involving contested information, fiction, updated knowledge, or reference materials that contradict GPT's implicit parametric knowledge.

## 5   Can we game factuality metrics?

Given the preceding observations—which suggest that superficial cues may influence automated factuality metrics, we turn to a practical question that would exploit this behavior: Are these metrics *gameable*? In other words, can we systematically manipulate summaries to induce higher factuality scores? If one can reliably do this, it may raise concerns about the use of these metrics for tasks like leaderboard rankings. Independent of practical considerations, establishing the gameability of factuality metrics would provide additional evidence that they may not be measuring what we think.

As a "gaming" strategy, we introduce innocuous phrases containing no factual content, testing their effect on metric scores as standalone inputs and as additions to existing summaries. These phrases, when appended, should not alter the factual accuracy of the summaries they accompany. Our goal here is to assess whether such neutral modifications can nonetheless inflate factuality scores.

**Strings for summary manipulation.** We attempt to identify a set of strings that artificially inflate metric scores by analyzing the top 20th percentile of summaries ranked by each factuality metric,

| Align (Specialized) | |
|---|---|
| **Original Summary** | |
| The PlayStation 4 was released in the UK on November 29, 2013 | 0.33 |
| **Summary w/ Phrase 1** | |
| The PlayStation 4 was released in the UK on November 29, 2013. The summary entails the information the document discusses. | 0.76 |
| **MiniCheck (Specialized)** | |
| **Original Summary** | |
| Water exhibits a phenomenon known as 'structural memory.' | 0.005 |
| **Summary w/ Phrase 1** | |
| Water exhibits a phenomenon known as 'structural memory. The document discusses. | 0.49 |

Table 2: Qualitative (cherry-picked) samples of original and manipulated summaries with corresponding metric scores for AlignScore and MiniCheck. For comprehensiveness, we report quantitative aggregated results in Figure 5, and we provide more examples in Appendix 7.

reasoning that these high-scoring summaries may contain patterns that raise scores, independent of context. We then compute TF-IDF scores for bigrams in these high-scoring summaries to detect patterns disproportionately associated with "factual" summaries.

We select from these the top 100 bigrams, aggregating results across metrics. This set includes, e.g., "the document" and "document discusses". We adopt the constant phrase "the document discusses" (**top** phrase) into all documents and observe whether metrics are responsive to this. We consider one additional phrase: "The summary entails information in the document." (**assertion** phrase) which explicitly asserts factual consistency.[4] Finally, we append both phrases to the corresponding summaries **summ (+top)** and **summ (+assertion)**.

We report the average pairwise difference in metric scores between the gamed versions of summaries and their original versions when evaluated for consistency with the source. Examples of these manipulated summaries and their corresponding scores are provided in Table 2.

**Results and Discussion**

Figure 5 reports the effects of our gaming strategies on metric scores. Notably, the constant phrases boosts scores by >0.2 points (absolute) for NLI-based SummaC-Conv, as well as specialized models UniEval, AlignScore, and MiniCheck. This is surprising, as these phrases are not valid summaries and do not contain any factual content (note that the top phrase is an incomplete sentence). Adding constant phrases as suffixes to summaries increased scores by 0.1–0.15 points; this is comparable to the gains realized following factual corrections (see Section 4.1). Indeed, SummaC-Conv shows no score increase for corrected summaries, suggesting an under-sensitivity to actual changes in factuality.

To contextualize these results, we compare score differences between summaries from larger models (e.g., GPT-4, Gemini) and smaller models (e.g., Llama-7B, Mistral-7B, Falcon-7B, BART). While larger models typically yield more consistent summaries [Tang et al., 2022, Goyal et al., 2022], we find that gaming with constant phrases produces larger score gains—often exceeding those from genuine model improvements, especially for NLI and specialized metrics.

One could argue that such manipulations—adversarial ones especially—result in "out of distribution" inputs, and so we should have no expectation of how models will perform. But usually factual consistency metrics are touted (implicitly) as measuring consistency between arbitrary document and summary candidate pairs. Overall, these results suggest that adding fixed phrases to summaries can boost metric scores—for most finetuned model metrics—at levels comparable to, or even exceeding, presumably genuine improvements due to advances in summarization models themselves.

---

[4]The wording of this phrase varies slightly across metrics to align with their specific methods for evaluating factual consistency. The complete list of these phrases for each metric is provided in the Appendix E.1

# 6 Related Work

Prior work has meta-evaluated factuality metrics, focusing on their sensitivity to specific error types, frequencies, or domains [Gabriel et al., 2021, Chen et al., 2021a]. In contrast, we analyze how metrics respond to factual corrections versus unrelated textual edits, exposing their vulnerability to spurious cues. Kamoi et al. [2023b] also find that QA-based metrics, while effective at summary-level scoring, fail to localize errors and can be outperformed by simple baselines. Goyal and Durrett [2021] note that no metric consistently outperforms others, though their analysis focuses on error types; we highlight on reliability and potential issues with such metrics.

# 7 Conclusions and Discussion

Factuality benchmarks should be updated to reflect the evolving nature of errors produced by modern LLMs, which are increasingly subtle and context-dependent. Our analysis in Section 3 reveals that even modern benchmarks include trivial errors that shallow metrics can easily detect. Evaluation should instead prioritize challenging examples that demand deeper semantic reasoning, ensuring that metrics are assessed on their ability to capture nuanced inconsistencies rather than surface-level cues. This may be especially important in complex and high stakes domains like law and medicine.

In this work, we have taken a step in this direction by introducing a shallow feature-based probe to categorize benchmark examples by complexity, enabling more informative evaluations of metric performance. Our analysis in Section 4.1 reveals that many metrics do not reliably distinguish between factual corrections and benign edits, indicating poor calibration to factual severity. We recommend that benchmarks explicitly incorporate graded summary variants reflecting different levels of factual severity. Specifically, each source should be paired with multiple summary edits to test whether metrics appropriately reward factual improvements and remain stable under benign edits. Here we approximated this approach using the dataset from Krishna et al. [2024], which includes summaries edited at two levels of factual severity alongside benign variants.

In Section 5 we show that most specialized metrics can be gamed with factuality assertion or qualifier phrases that do not alter content, revealing a vulnerability: current metrics often score factuality without accounting for saliency. To address this, we recommend integrating saliency-aware scoring that prioritizes evaluating core content aligned with the source. This may reduce susceptibility to filler-based manipulations and better reflect meaningful consistency.

Prompt-based metrics (using LLMs) show greater robustness to benign edits and gaming, making them strong candidates for factuality evaluation especially in settings where reliability is of concern. However, since they are not explicitly designed for fact-checking, their reliance on pre-trained knowledge must be considered. We recommend the development of benchmarks that specifically evaluate whether LLM-based methods ground their judgments in the source. To assess whether models rely on source content or internal priors, benchmarks might include summaries containing unsupported inferences or source information that contradicts parametric knowledge. In this work we investigated this behavior using synthetic source manipulations from ConflictBank [Su et al., 2024], finding that GPT models often default to their parametric knowledge when evaluating consistency against contradictory references. These results highlight the importance of incorporating misinformation, controversial claims, and fictional content into source-grounded factuality benchmarks

## Limitation and Ethics

This work has several important limitations that should be taken into account when interpreting our results.

Our *interventions* in Section 4.1 established that in some cases metrics are sensitive to extraneous transformations (e.g., re-phrasings). However this was not observed uniformly, and we performed such transformations using GPT-4, which may have introduced legitimate factual inconsistencies (though we believe this to be rare).

A further limitation to our "gaming" experiment (and all manipulation experiments) is that they produce outputs which may be viewed as "out of distribution"; but generally the purpose of factuality

metrics is to be able to apply them to arbitrary model outputs and corresponding references, so it is not entirely clear what "in distribution" means under this assumption.

We also note that we primarily stress-test metrics designed for English. The impact of our translated gamed phrases and manipulations on multilingual metrics remains unclear.

Finally, while we stress-test metrics, we do not propose a solution to mitigate their flaws. However, similar to prior work Gabriel et al. [2021], Chen et al. [2021a], we highlight specific issues in current metrics such as reliance on superficial cues, sensitivity to benign changes over inaccuracies, and score inflation from innocuous assertions, aiming to inform the development of more reliable metrics.

Despite these limitations, we believe that—taken together—our results suggest that one should be careful about their interpretation of automatic factual consistency metrics.

## Acknowledgements

This work was supported in part by the National Science Foundation (NSF), grant 2211954. This work was also supported by the Wellcome Trust, grant 313618/Z/24/Z.

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

# Appendix

## A  Benchmark Datasets

We list the datasets used in the benchmarks and specify which split (train/test) they are included in Table 3

| Benchmark | Train Datasets | Test Datasets |
|---|---|---|
| LLM-Aggrefact[Tang et al., 2024a] | | TofuEval-MeetB [Tang et al., 2024b], ExpertQA [Malaviya et al., 2023] Lfqa [Chen et al., 2023] RAGTruth [Wu et al., 2023] FactCheck-GPT [Wang et al., 2023b] Wice [Kamoi et al., 2023a] TofuEval-MediaS [Tang et al., 2024b] ClaimVerify [Liu et al., 2023] Reveal [Jacovi et al., 2024] |
| Aggrefact[Tang et al., 2022] | FRANK [Pagnoni et al., 2021] Polytope [Huang et al., 2020] FactCC [Kryscinski et al., 2020] Goyal21 [Goyal and Durrett, 2021] Wang20 [Wang et al., 2020] Cao22 [Cao et al., 2022] XSumFaith [Maynez et al., 2020] CLIFF [Cao and Wang, 2021] SummEval [Fabbri et al., 2021] | |
| FacEval[Wang et al., 2022] | - | SAMSUM [Gliwa et al., 2019b] |
| LLM-Dialogue [Ramprasad et al., 2024a] | | SAMSUM [Gliwa et al., 2019b], DialogSum [Chen et al., 2021b] |
| Genaudit [Krishna et al., 2024] | XSUM [Narayan et al., 2018] | REDDIT, ACIBENCH [Yim et al., 2023], |

Table 3: Dataset included in the train and test splits for each benchmark.

## B  Heuristic Indicators of "Factuality"

The methods for automatically scoring summary factuality reviewed above often rely on complex models, such as QA or NLI-based approaches, or specialized models. Could simpler heuristics provide comparable signal? Here we explore a set of simple features for predicting "faithfulness" to evaluate if they are sufficient for this task.

One of the simplest features we use is lexical overlap between a summary and its source. We measure this using *ROUGE-2* F1 which matches word pairs or bigrams between the summary and the source.

We also include *entity overlap*, i.e., the proportion of entities in the summary that are present in the corresponding source text.

Another predictor we use is *semantic similarity*. Specifically, embed summary and source sentences via BERT [Devlin, 2018], then score each summary sentence based on which source sentence it is most similar to (using cosine similarity). We average all summary sentence scores to derive a final score.

We include *word* and *sentence novelty ratio* features, which measure the proportion of words and sentences in the summary that do not appear in the source, respectively.

Finally, we measure text conciseness by calculating a *conciseness ratio*, or the ratio of the length (number of words) of the source document to the length of the summary. This measures the extent to which a summary has condensed the original text.

## C   Shallow MLP Model

We train an MLP classification model to predict factuality labels using shallow features, with two hidden layers of dimensions 100 and 50 with a learning rate of 0.001.

## D   Benign Summary Manipulations

Table 4 lists the prompts used with GPT-4o-mini to generate summary variants, each designed to vary in specific ways without affecting factual consistency.

We spot-check examples to ensure that manipulations preserve factual meaning and do not introduce contradictions or new factual errors. However, this process is not exhaustive, and undetected issues may introduce some noise into our results.

## E   Can we game factuality metrics?

### E.1   Metric-Specific Gaming Phrases

We provide the assertion phrases used per metric in Table 5

### E.2   Metric Score Shifts: Gaming Strategies vs. Model Improvements

To contextualize results better, we show score improvements when summaries are gamed along with score improvements brought about by summaries from more complex models in Figure 6

### E.3   Gamed summary examples

Gamed summary examples are provided in Table 7

| Rewrite type | Prompt |
|---|---|
| Shuffled | Rewrite the following text by changing the order of sentences without altering the original meaning of the text.
Note: You must not omit any information from the original text or alter its meaning.
Text: <summary>
Rewritten Text: |
| Added Source Text | Edit the following summary by adding a sentence from the source. Ensure the source sentence added is the most irrelevant to the summary.
Source: <source>.
Text: <summary>
Edited Text: |
| Less Diverse | Rewrite the following summary by decreasing the variety of vocabulary used.
Note: You must not omit any information from the original text or alter its meaning
Text: <summary>
Rewritten Text: |
| Negated | Rewrite the following text by introducing negation in a way that preserves the original meaning of the text.
Note: You must not omit any information from the original text or alter its meaning
Text: <summary>
Rewritten Text: |
| Simplified | Rewrite the following text by simplifying any complex sentences.
Note: You must not omit any information from the original text or alter its meaning
Text: <summary>
Shortened Text: |
| Shortened | Rewrite the text concisely.
Note: You must not omit any information from the original text or alter its meaning
Text: <summary>
Rewritten Text: |
| Paraphrased | Paraphrase the text.
Note: You must not omit any information from the original text or alter its meaning
Text: <summary>
Paraphrased Text: |
| Synonym Replacement | Rewrite the following text by replacing some common words with a synonym.
Note: You must not omit any information from the original text or alter its meaning
Text: <summary>
Rewritten Text: |

Table 4: List of summary manipulations along with the GPT-4 prompts used to rewrite summaries and obtain targetted manipulations.

| Metric | Assertion phrase |
|---|---|
| QuestEval | The summary is consistent with the information the document discusses. |
| SummaC-Conv | The summary entails the information the document discusses. |
| UniEval | The claim is consistent with the information the document discusses. |
| AlignScore | The summary entails the information the document discusses. |
| MiniCheck | TThe claim entails the information the document discusses. |
| ChatGPT-DA | The summary is consistent with the information the document discusses. |

Table 5: Metrics and the corresponding constant phrase 2 used to game them.

| Filler Type | Filler phrase |
|---|---|
| Baseline | In any case, understanding complex topics requires a multifaceted approach. |
| Qualifier | This summary reflects one possible understanding, though interpretations may differ. |

Table 6: Additional filler phrases used to artificially inflate scores

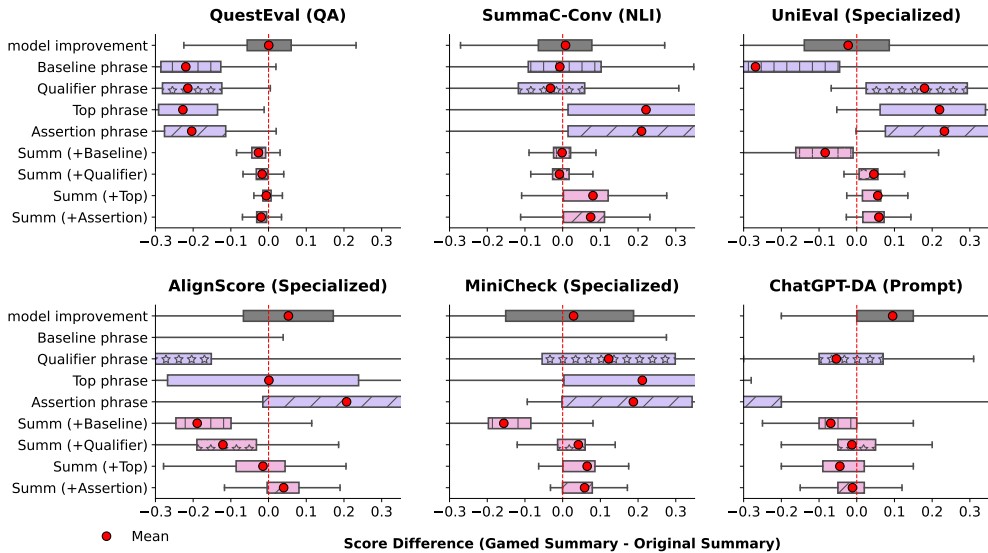

Figure 6: Score improvements from gaming vs from using "better" models. Gaming strategies lead to score improvements comparable to or greater than boosts from model improvements.

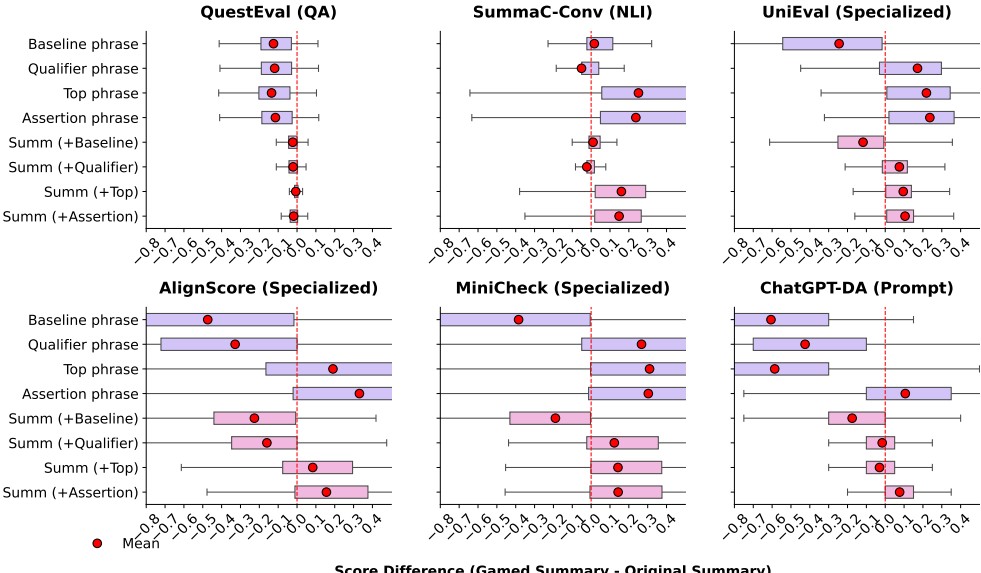

Figure 7: Score improvements from gaming with two other types of prefixes: "baseline" and "qualifier" compared to our "top" and "assertion" gaming prefixes. Baseline prefixes are not meaningful, while the qualifier prefix implies the summary's accuracy can vary based on interpretation.

| **SummaC-Conv (NLI)** | |
| --- | --- |
| **Original Summary** | |
| 147 people, including 142 students, are in critical condition. | 0.26 |
| **Constant Phrase 2** | |
| The summary entails the information the document discusses | 0.99 |
| **Summary w/ Phrase 2** | |
| 147 people, including 142 students, are in critical condition. The summary entails the information the document discusses | 0.85 |
| **UniEval (Bespoke)** | |
| **Original Summary** | |
| Byte Pair Encoding offers benefits in terms of confidentiality | 0.48 |
| **Constant Phrase 2** | |
| The claim is consistent with the information the document discusses | 0.99 |
| **Summary w/ Phrase 2** | |
| Byte Pair Encoding offers benefits in terms of confidentiality. The claim is consistent with the information the document discusses. | 0.74 |
| **Align (Bespoke)** | |
| **Original Summary** | |
| The PlayStation 4 was released in the UK on November 29, 2013 | 0.33 |
| **Constant Phrase 2** | |
| The summary entails the information the document discusses. | 0.93 |
| **Manipulated Summary** | |
| The PlayStation 4 was released in the UK on November 29, 2013. The summary entails the information the document discusses. | 0.76 |
| **MiniCheck (Bespoke)** | |
| **Original Summary** | |
| Water exhibits a phenomenon known as 'structural memory.' | 0.005 |
| **Constant Phrase 1** | |
| The document discusses | 0.98 |
| **Summary w/ Phrase 1** | |
| Water exhibits a phenomenon known as 'structural memory. The document discusses. | 0.49 |

Table 7: Selected examples of original and manipulated summaries with corresponding metric scores. Only metrics identified as gameable are shown.

