# OpenReview forum: "Do Automatic Factuality Metrics Measure Factuality? A Critical Evaluation"
_NeurIPS.cc/2025/Conference — NeurIPS 2025 poster_

### Official Review · Reviewer_SyLu · 2025-06-26

**Clarity:** 4
**Significance:** 3
**Originality:** 3
**Rating:** 5
**Confidence:** 4

**Summary:**

This paper evaluates the extent to which commonly used factuality (faithfulness) metrics genuinely reflect the factual consistency of generated outputs. It also benchmarks the robustness of these metrics against a range of perturbations. The authors find that many existing metrics are vulnerable to manipulation, and that ChatGPT-based metrics often rely more on parametric knowledge than on the provided source.

**Questions:**

See weaknesses.

**Ethical Concerns:**

["NO or VERY MINOR ethics concerns only"]

**Final Justification:**

After reading the authors' rebuttal and the other reviews, including the more critical ones such as reviewer kyR2's, I feel confirmed in my understanding and assessment of the paper. It is reassuring to see that several reviewers share similar views on the strengths of the work, even when they assign a lower score.

While some of the concerns raised are valid and worth noting, I do not believe they significantly affect the overall quality or contribution of the paper, as even the critical reviews acknowledge.

I maintain my score and express my support for accepting the paper.

**Limitations:**

Yes

**Paper Formatting Concerns:**

It looks like the citations are not formatted correctly (it should be numbered and not with the authors names).

**Quality:**

4

**Strengths And Weaknesses:**

**Strengths**

- The paper is well written and highly engaging.

- It addresses a timely and important problem in the evaluation of text generation.

- The claims are clearly supported by well-designed and rigorous experiments.


**Weaknesses**

- As acknowledged by the authors, Figure 4(b) presents weak evidence and is not fully convincing.

- The use of GPT-4 for generating edits, while pragmatic, could introduce bias given that one of the evaluated metrics is also based on GPT-4.

- The “easy / medium / hard” classification of summaries would benefit from additional empirical examples or clarification.

- A more detailed discussion of the human-annotated datasets used for metric evaluation could strengthen the presentation.

---

> ### Author Rebuttal · Authors · 2025-07-31
>
> We thank the reviewer for their thoughtful comments about the work. We address specific issues raised below.
>
> **Re: "The use of GPT-4 for generating edits, while pragmatic, could introduce bias given that one of the evaluated metrics is also based on GPT-4"**
>
> This is a fair point, and we will note it in the revised paper. However, we note that ChatGPT-DA uses a prompt explicitly asking for an assessment of factual consistency between a summary and source, whereas the edits we introduce with GPT-4 are targeted edits explicitly requested: We think it is unlikely that the ChatGPT-DA metric is somehow advantaged because GPT-4 was used to make these edits, though it is possible. (This is not like a scenario where we are simply asking GPT which of two generated examples it “prefers”, where one has been generated with GPT.)
>
> We also note that if this did confer an advantage to ChatGPT-DA, it would in a sense make the results in 5.2 even more compelling, since we found that the metric is not reliably responsive to inconsistencies being introduced into summaries (these inconsistencies were elicited from GPT-4, so it “knows” they are inconsistent).
>
> **Re: "The “easy / medium / hard” classification of summaries would benefit from additional empirical examples or clarification."**
> We agree with the reviewer. We will provide additional clarification around these categories and we will include illustrative examples from each category in the Appendix.
> Regarding clarification about these categories: The idea here is that some examples can be readily classified as “factual” (or not) using shallow features like word overlap; these are “easy” examples. By contrast, “medium” and “hard” categories capture examples that are difficult to classify with respect to factuality using these shallow features. We operationalize these categories as follows. “Easy” instances are those that the MLP predicts correctly with high confidence (these are examples that are easy to get right using only shallow features). We consider “medium” examples to be those where the MLP is again correct, but where it has lower confidence. Finally, “hard” instances are those that the MLP misclassifies. We will elaborate this in the text.
>
> **Re: "A more detailed discussion of the human-annotated datasets used for metric evaluation could strengthen the presentation."**
>
> We agree that this would make the work more self contained (at present these details are in the original papers describing the corresponding corpora). We will add a section to the Appendix that discusses in greater detail the annotated datasets we have used, and point to this in the main text for the interested reader.

---

### Official Review · Reviewer_pjyy · 2025-06-29

**Clarity:** 3
**Significance:** 2
**Originality:** 3
**Rating:** 5
**Confidence:** 3

**Summary:**

I appreciate the authors for running a pretty extensive set of comparisons of so-called factuality check methods along many different aspects, such as sensitivity to surface-level features, sensitivity to fact-invariant revisions and adversarial edits. An overall finding at the surface level (pun intended) is that none of the proposed approaches is as good as they were claimed to be originally. Although the authors constantly conclude that prompt-based metrics, such as ChatGPT-DA and MiniCheck, are reasonably good and better than the other metrics, my interpretation is that every metric has some weaknesses that make them not adequate to be used for checking the factuality of any textual statement.

Despite this weakness (in the authors’ interpretation rather than their experimentation), I do believe it is good for the whole community to have this paper. This paper tells us about the limitations of existing factuality benchmarks (that said, I do not believe there could be a “standard” benchmark for factuality, since factuality is almost always in the eyes of beholders,) and that different metrics have different weaknesses and strengths.

The only thing i miss from this submission is the discussion on factuality itself. Is factuality something we can build a benchmark and test language generators for? Should we approach it as a leaderboard chasing game, when factuality is not about the average behaviour? I would suggest the authors remove Fig. 4(b) (completely uninformative) and add one paragraph on this discussion.

**Questions:**

See my summary above.

**Ethical Concerns:**

["NO or VERY MINOR ethics concerns only"]

**Limitations:**

See my summary above.

**Paper Formatting Concerns:**

No concern

**Quality:**

3

**Strengths And Weaknesses:**

See my summary above.

---

> ### Author Rebuttal · Authors · 2025-07-31
>
> We thank the reviewer for their thoughtful feedback and interpretation of the work. We agree entirely with the reviewer’s perspective: Every “factuality” metric has certain weaknesses, and none should be taken as a sort of “gold standard”. Our hope is that this work elucidates the issues (and strengths) particular to each kind of factuality metric, ultimately to enable more informed use and interpretation of such metrics, when appropriate.
> We also agree that the work would benefit from a broad discussion of “factuality” and its measurement. One of the motivations for the “gameability” experiments (Section 6) was to highlight that if we “leaderboard” chase “factuality”, this may lead to researchers exploiting weaknesses in metrics, which we have shown is possible. That is, “factuality” metrics may fall victim to Goodhart’s Law, which says that a metric ceases to be useful as soon as it is optimized for. On the other hand, given the importance of factuality, we do not think its measurement should be abandoned entirely; only that practitioners should be aware of the limitations of automated metrics for something as slippery as “factuality”. We will add a paragraph discussing these issues to the paper, as suggested by the reviewer.

---

### Official Review · Reviewer_o3Me · 2025-06-29

**Clarity:** 2
**Significance:** 2
**Originality:** 2
**Rating:** 3
**Confidence:** 4

**Summary:**

The paper stress tests multiple factuality metrics to determine their in depth meaning. The authors finds that all metrics struggle in hard examples, many metrics are sensitive to benign edits, some metrics can be gamed by adding irrelevant phrase, and GPT relies on internal knowledge rather than the provided documents/context.

**Questions:**

1. Do the gaming strategies work across different domains and text types, or are they specific to the datasets tested?
2. Are the reported differences between metrics statistically significant? The paper mentions confidence intervals in Section 5.2 but statistical testing across all experiments is unclear.
3. I would like to see the explanation for the questions raised in the weakness part.

**Ethical Concerns:**

["NO or VERY MINOR ethics concerns only"]

**Final Justification:**

The rebuttal improves the clarity of several methodological components and helps contextualize the results more effectively.

**Limitations:**

Yes

**Quality:**

3

**Strengths And Weaknesses:**

**Strengths**
1. The evaluation includes multiple datasets and covers diverse domains.
2. Gaming experiments effectively demonstrate the vulnerability of existing metrics to superficial manipulations which reveals important limitations in current evaluation approaches.

**Weakness**
1. Gaming idea is very impressive but I was wondering how this matrics varied the score so much given the phrases ("the document discusses", "the summary entails.) which are very impractical. While the gaming results are impressive, the extreme score improvements (e.g., from 0.005 to 0.49 for MiniCheck) seem suspiciously large. The TF-IDF setup for identifying gaming phrases needs much clearer explanation. How exactly were these phrases selected, and why do they produce such dramatic effects?
2. The paper uses a simple MLP with basic features to categorize "easy" vs "hard" examples. This classifier achieves modest performance, raising questions about whether the difficulty categorization is meaningful.
3.No code is provided, making it extremely difficult to understand the experimental setup. This significantly undermines the paper's impact and verifiability as it is hard to understand the all the setup from given prompts only. Many experimental procedures lack sufficient detail for replication.
4. Line 155 describes the difficulty categorization in a very confusing manner.
5. The relationship between prediction accuracy, confidence, and difficulty labels is unclear.
6. The shallow MLP baseline methodology needs better explanation.
7. KeyBERT is mentioned but its usage context is unclear.
8. Appendix B mentions "conciseness ratio" but doesn't define it clearly.
9. Typos

- Figure 1 upper right: "me" should be "m̂" (m hat)
- Line 211: "Tu" should be "To"
- Line 554: "GPT-40" should be "GPT-4o"
- Line 561: "summer" should be "summary"
- Line 92: Missing full stop
- Line 122: Awkward phrasing
- Line 560: Missing full stop after "Table 5"
- Line 575: Missing full stop

---

> ### Author Rebuttal · Authors · 2025-07-31
>
> We thank the reviewer for their comments and respond to the specific issues raised below.
>
> **Re: "Gaming idea is very impressive but I was wondering how this metrics varied the score so much given the phrases ("the document discusses", "the summary entails.) which are very impractical..."**
>
> We agree with the reviewer that these are surprisingly large swings, and indeed find the exercise interesting for this reason. We also note that, as noted in the caption of Table 2, the particular change cited (0.005 to 0.49) was a specific qualitative example where the inclusion of one of the phrases (phrase 1) substantially changed the score. The average changes for MiniCheck over all examples (reported in Figure 5) ranges from 0.15 to about 0.30; still large but not quite as dramatic.
>
> We agree with the reviewer that the procedure we used for identifying phrases could be clarified, and we will edit the paper to better describe this procedure. Briefly, we compute TF-IDF scores for bigrams across documents and manually examine the highest-scoring bigrams that are disproportionately associated with summaries labeled as ``factual'' i.e. summaries with high consistency scores. We emphasize that the main aim here was to show that such phrases exist, and this simple approach demonstrably accomplished this aim (since we identify phrases that do consistently influence results). We acknowledge that more sophisticated methods for searching for such phrases may find even more effective strings .
>
> **Re: "The paper uses a simple MLP with basic features to categorize "easy" vs "hard" examples. This classifier achieves modest performance, raising questions about whether the difficulty categorization is meaningful..."**
>
> We agree that the paper should better motivate our aim with this set of experiments and the construction of the MLP, and also that our description of difficulty categorization as written is confusing.
>
> In brief, we intentionally use a “simple” model here (i.e., one based on “shallow” indicators like word overlap) to try and identify cases where it is relatively easy (and maybe even trivial) to spot (in)accurate summaries in the sense that an MLP using only surface-level features can do so with high confidence. If all examples were this straightforward, we would not need sophisticated factuality scoring models, because a simple model would suffice. But there are instances where the simple MLP does not do as well, e.g., those in the medium and hard categories. Our objective here is to understand how metric performance varies as the complexity of the evaluation task increases and use the "simple" model (MLP’s) limitations as a proxy to surface examples that require more nuanced evaluation.
>
> We would argue that the results in Figure 2 provide strong evidence that the difficulty categorization is meaningful: The average metric performance declines by about 21 absolute points on “easy” vs “hard” examples, according to the MLP categorization. (The decline is about 11 points from “easy” to “medium”.)
> Regarding these categories and the reviewer’s comment about the confusing description on L154-155: The idea here is that some examples can be readily classified as “factual” (or not) using shallow features like word overlap; these are “easy” examples. By contrast,  “medium” and “hard” categories capture examples that are difficult to classify with respect to factuality using these shallow features. We operationalize these categories as follows. “Easy” instances are those that the MLP predicts correctly with high confidence (these are examples that are easy to get right using only shallow features). We consider “medium” examples to be those where the MLP is again correct, but where it has lower confidence. Finally, “hard” instances are those that the MLP misclassifies. We will clarify this in the text.
> We also agree with the reviewer that we should add more details about the MLP and its performance; we will do so in the updated draft, and also add more text explaining our motivation here.
>
> **Re: "No code is provided, making it extremely difficult to understand the experimental setup. "**
>
> We fully agree with the reviewer that the code should be available, and will release code and data to reproduce all results. We apologize for not making an anonymized repository available ahead of submission; we would do so now, but the program chairs have explicitly forbidden it, unfortunately (they “prohibit using any links in the rebuttal, including but not limited to anonymous or non-anonymous URL links”).
>
> **Re: "KeyBERT is mentioned but its usage context is unclear."**
>
> We agree that this should be clarified in the paper. Briefly, KeyBERT is an existing method for extracting keywords from a set of documents. We used this to find key terms in source documents that we could use to search over large pretraining corpora (Dolma v1.6, the pretraining data used for OLMo models). The idea is to investigate a relationship between topic (keyword) frequency in pretraining and responsiveness to edits. The hypothesis here is that models might be more responsive to contradictions introduced in documents on rare topics, because it has comparatively little parametric of these. We find weak evidence supporting this in Figure 4(b).  We will elaborate on KeyBERT and its usage here in the paper.
>
> **Re: "Appendix B mentions "conciseness ratio" but doesn't define it clearly."**
>
> We apologize for any ambiguity here. This is merely the number of words in the source documents divided by the length of the summary. The bigger this is, the more the summary has compressed the source in terms of length.
>
> We also thank the reviewer for pointing out various typos and we will fix all of these in revision.

---

> > ### Comment · Reviewer_o3Me · 2025-08-05
> >
> > I appreciate the stated intent to release code and data, though the lack of availability during review does limit the ability to fully assess the experimental pipeline. The rebuttal improves the clarity of several methodological components and helps contextualize the results more effectively. I have increased my clarity and quality scores in response to the authors' explanations, but I stand by my original overall rating.

---

### Official Review · Reviewer_kyR2 · 2025-07-01

**Clarity:** 3
**Significance:** 2
**Originality:** 3
**Rating:** 3
**Confidence:** 4

**Summary:**

This paper presents a comprehensive evaluation of model-based factuality evals, focusing on whether a model-generated summary is grounded in the source document. The paper presents a series of experiments with interesting findings, e.g., that the correctness of these evals correlates strongly with difficulty levels defined by spurious features (e.g., word overlap with the source document), that model-based evals are sensitive to benign modifications while being insensitive to meaningful modifications (e.g. contradictions), that LLM-based evals rely heavily on their internal knowledge rather than the source document, and that these LLM-based evals are gameable, e.g., by adding a sentence like “The summary entails the information the document discusses”.

**Questions:**

Would appreciate authors' thoughts on the weaknesses in the previous field.

**Ethical Concerns:**

["NO or VERY MINOR ethics concerns only"]

**Quality:**

3

**Strengths And Weaknesses:**

Strengths
- The paper evaluates a wide range of factuality evals and includes diverse summarization benchmarks, capturing a broad range of error types and settings.
- The idea of using spurious features like word overlap and entity overlap to define and categorize difficult levels is useful.
- The analysis with different types of perturbations such as corrections and paraphrases is insightful.
- I found the recommendation at the end of the paper particularly insightful, such as cautions against using model-based evals for text that is counterfactual or does not align with LLMs’ internal knowledge such as myths, misinformation, or uncommon facts.

Weaknesses

I do not believe there are particular flaws in the paper. I do have some reservations about certain conclusions drawn from the experiments. For instance:
- For the first experiment in Section 5.2, I do not believe that little gap between evaluations with and without access to the source document necessarily indicates that the model relies heavily on parametric knowledge. If the content of the source document is already aligned with a model’s internal knowledge, the gap may naturally be small, even when the evaluator does ground to the source document when it has access to it.
- The second experiment in Section 5.2 is more compelling. However, the nature of counterfactuals in the modified text can affect the result, e.g., text that is intentionally written to be contradictory to the original text is likely to be highly unrealistic and unnatural. It would have been more interesting if the text is realistically counterfactual, e.g., myth, or text about very recent events that the LLMs would not have internal knowledge about.
- Section 6 feels less compelling to me. There’re always ways to make adversarial examples to fool the evaluator, e.g., adding a sentence “The summary entails the information the document discusses” looks like a type of a jailbreaking attack. However, this is unlikely to be actually generated by a reasonable summarization model. If a takeaway is that a summarization model shouldn’t be trained with these metrics (e.g., as a reward model) because it’s too vulnerable to reward hacking, I think it is reasonable. However, assuming that the summarization model is not directly exploiting this metric, it is unclear how relevant this finding is in real world scenarios.

I also think the overall findings of the paper are, while solid, not particularly surprising or new. It is already well understood that model based evaluators are far from perfect, but the reason we still use them is because they are the only practical & cost-efficient alternative to human evaluation. And I do not believe that this paper is offering actionable strategies for improving these model-based evaluations.

I particularly appreciated the recommendation to exercise caution when applying model-based factuality evals in text involving myths, misinformation, or uncommon facts, and think it would have been significantly better if the paper had directly demonstrated this issue. This paper took a different route by making various synthetic perturbations, which is valuable, but I think this always raises questions on how well these findings on synthetic text generalize to real-world text.

Minor questions/suggestions:
- The categorization of factuality evaluation methods into QA-based, NLI-based, and LLM-based does not seem to be most informative in my opinion, e.g., QA vs. NLI is a matter of a format choice rather than a fundamental methodological difference. How about prompt-based vs. fine-tuning based (which could either be QA or NLI)?
- Figure 2: The scales across three graphs appear inconsistent, which makes visual comparison difficult. In addition, I think it might be more effective to group results by eval methods and compare performance across difficulty levels within each group, rather than grouping by difficult levels and comparing across eval methods.

---

> ### Author Rebuttal · Authors · 2025-07-31
>
> We thank the reviewer for their thoughtful feedback and are glad that they found the analyses interesting. It seems that the reviewer appreciated the findings here but has some reservations about the conclusions drawn and how we have presented them; we appreciate their comments  and feel we can readily address this in revision by adding additional context around the results presented.
>
> **Re "For the first experiment in Section 5.2, I do not believe that little gap between evaluations with and without access to the source document necessarily indicates that the model relies heavily on parametric knowledge..."**
>
> We agree that this does not necessarily indicate that the model relies heavily on parametric knowledge; it is, however, a piece of evidence suggesting that it might. We ran the second experiment exactly because this first exercise is merely suggestive, and are glad the reviewer found this more compelling. That said, we will add a comment to our description of the first experiment in 5.2 noting that—as the reviewer rightly points out—there is an alternative plausible explanation for this particular initial finding (i.e., that the source documents are aligned with model parametric knowledge already).
>
> **Re: "The second experiment in Section 5.2 is more compelling. However, the nature of counterfactuals in the modified text can affect the result."**
>
> This was a targeted experiment intended to deepen the preceding analysis, as suggested by the reviewer. These are focussed edits that we made in a way that preserves readability and fluency; these are minimal rewrites. Therefore, we would not characterize these as “highly unrealistic and unnatural”, though it is true that they are—by construction—synthetic.
>
> **Re: "It would have been more interesting if the text is realistically counterfactual eg., myth, or text about very recent events that the LLMs would not have internal knowledge about."**
>
> We agree that this is very interesting, and have conducted a set of experiments based on this suggestion which we will add to the paper.
>
> Consistent with our guidance, the punchline is: LLM-based consistency evaluation fares worse when it is presented with summaries to be  scored with respect to inputs that themselves contain counterfactual (factually incorrect) information. This experiment (which we elaborate on below) strengthens the case for being careful of using LLM based metrics for consistency check when evaluating summaries of mythological documents or documents that contradict GPT's internal knowledge.
>
> The details of the experiment we ran, which we will add to the paper, are as follows. We used the “ConflictBank” dataset (Su et al. 2024), which comprises factual claims taken from Wikipedia along with corrupted non-factual claims obtained by performing targeted substitutions in the factual claims. Each non-factual claim is associated with a corrupted reference that supports it—that is, this corrupted reference is also non-factual in a way that supports the non-factual claim. For each factual claim, we obtain corresponding factual (uncorrupted) references from Wikipedia. So we have four pairs:
> (a)  <factual articles, factual summaries>,
> (b) <counterfactual (non-factual) articles, counterfactual (non-factual) summaries>,
> (c) <factual articles; counterfactual (non-factual) summaries>, and
> (d) <counterfactual (non-factual) articles, factual summaries>.
>
> From a consistency evaluation perspective, (a) and (b) are equally “consistent” (both are cases where the summaries are consistent with the inputs), and (c) and (d) are equally “inconsistent” (both are cases where the summaries contradict the inputs). The only difference is that in (c), the summaries are factually incorrect—counterfactual—with respect to the world at large (and potentially knowledge that GPT has learned), while in (d) they are factually accurate (even while contradicting the corresponding inputs).
> We aim to evaluate whether LLM-based factuality scoring is compromised when evaluating content against counterfactual references, even when those references support the claims being evaluated. To do so, we prompt GPT to score both supported/unsupported summaries against different reference text.
>
> In the counterfactual reference setting, we evaluate supported summary (b) and unsupported summary (d) against counterfactual references. Expected behavior would show b>d (supported summaries score higher). However, if d>b, this could suggest GPT's internal factual knowledge is overriding the reference-based evaluation, inappropriately favoring factually accurate but reference-unsupported content. We find that GPT incorrectly rates the unsupported (but factual) summary higher than the supported summary in 3% of cases.
>
> As a baseline, we also evaluate supported summary (a) and unsupported summary (c) against factual references. Here, GPT rates unsupported summaries higher only 0.2% of the time. The increase in incorrect summaries being rated higher (0.2% --> 3%) demonstrates that GPT struggles significantly more with consistency evaluation when references contain misinformation or contradict established facts. This suggests the model's evaluation reliability could be compromised in real-world scenarios involving contested information, updated knowledge, or reference materials that contradict GPT's knowledge.
>
> We also compare the score differences between supported and unsupported summaries in both settings, namely (a - c) and  (b - d), using a paired t-test on the two deltas. The test yields a t-value of 10.28 and p <<  0.05, i.e., the score gap between supported and unsupported summaries is significantly smaller when evaluated against counterfactual references compared to factual ones. This suggests that GPT struggles more to distinguish between supported and unsupported summaries when the reference contradicts its internal knowledge. In particular, when evaluating summaries against references that contradicts its internal knowledge, the model shows reduced sensitivity to whether the summary is actually supported by evidence.
> Again, we will add this result to the paper and thank the reviewer for the suggestion.
>
>
> **Re: "Section 6 feels less compelling to me. There’re always ways to make adversarial examples to fool the evaluator...it is unclear how relevant this finding is in real world scenarios."**
>
> We agree with the reviewer that we could do a better job contextualizing these results and discussing their potential implications. We will do so in the revised draft.
> Briefly, there are a few practical takeaways here in our view. First and foremost, ideal metrics are not “hackable” (e.g., in classification tasks, one cannot artificially inflate accuracy; even ROUGE is difficult to “game”, which made it a useful though imperfect metric for many years); adversarial approaches work here only because an LLM is being used as an assessor, which is a relatively recent innovation. As a direct implication of this, we agree with what the reviewer says: The fact that one can trivially inflate “factuality” scores in this way implies that using this as a reward signal in fine-tuning is probably a bad idea. So this is one practical takeaway which we believe to be relevant, and we will note this explicitly in the paper.
> Another practical takeaway is that such metrics alone are not suitable to compare models for factuality for, e.g., in “leaderboard” type setups (such as Vectara’s Hallucination Leaderboard) where researchers compete to build more “factually accurate” models, because one could ascend the leaderboard by including adversarial phrases to boost the LLM factuality score. Of course, additional rules could be put in place to try and disallow such “hacks”, but this might be harder than it appears: We have shown the existence of a few adversarial manipulations that inflate scores, but there are likely many such modifications that one could make. Some of these may be less obvious than the ones we used here. Factuality as a metric may then fall victim to Goodhart's Law (which states that a metric ceases to be useful as soon as it is optimized for), with researchers finding certain prefixes that consistently raise scores instead of meaningfully improving summaries.
>
>
> **Re: "The categorization of factuality evaluation methods into QA-based, NLI-based, and LLM-based does not seem to be most informative in my opinion..."**
>
> We think it is worth distinguishing between QA and NLI, but agree with the reviewer that the bigger differentiator is prompt-based vs. fine-tuned. We will revise the presentation to emphasis this split, while still reporting granular results to tease apart QA and NLI methods
>
> **Re: "Figure 2: The scales across three graphs appear inconsistent, which makes visual comparison difficult. In addition, I think it might be more effective to group results by eval methods and compare performance across difficulty levels within each group"**
>
> The reviewer is correct about the scales: We will fix this presentation issue in revision and appreciate the suggestion.
>
> We also welcome the grouping suggestion, and will add a plot presenting results this way as well.

---

> > ### Comment · Reviewer_kyR2 · 2025-08-04
> >
> > Thank you for your detailed responses - I appreciate the follow-up discussion! That said, I stand by my original review. From a technical perspective, the same argument can apply to any LLM-as-a-judge type of evaluation: we know these metrics are imperfect, but still use them because they're still the best available alternative to human evaluation. (For example, the authors mentioned ROUGE-L in the rebuttal - I’m genuinely curious whether LLM-based evaluations actually show lower correlation with human judgments than ROUGE-L in real-world model development settings (not the adversarial settings).) And I don't think we were using these metrics for leaderboard hill climbing either - they have been used mainly as a useful intermediate evaluation.
> >
> > This paper definitely raises a useful cautionary note, but I'm not sure if this is something the community wasn't aware of. Anyways, I've also read the other reviews, and it seems there's general agreement on the strengths and limitations of the work, and this will likely come down to AC's judgement!

---

> ### Author Response · Authors · 2025-08-04
> **Response to Reviewer kyR2**
>
> We thank the reviewer for their continued engagement! We offer some points of clarification:
>
> 1. We note that LLM-as-a-judge type evaluations are only one type of "factuality" metric that we have investigated in this work.
>
> 2. We agree with the reviewer that these metrics are often the best available alternative to human evaluation, and we are not saying that these metrics should not be used *at all*; instead we hope this work allows researchers and practitioners to use these with better understanding of their strengths and weaknesses. Many of these metrics generate continuous scores but there is limited understanding of what these scores actually represent. We seek to determine how accurately these scores reflect genuine "factuality" improvements and assess their robustness to confounding factors.
>
> 3. Past work (e.g., Mayznez et al. "On Faithfulness and Factuality in Abstractive Summarization" and Goyal et al "News Summarization and Evaluation in the Era of GPT-3") has shown that ROUGE fares worse on factuality evaluation than metrics explicitly crafted to measure this (like those we evaluate in this work).
>
> 4. In fact, there are leaderboards using factuality metrics, e.g., Hughes Hallucination Evaluation Model (HHEM) Leaderboard, available on HuggingFace. Beyond formal leaderboards, existing works depend on these metrics to assess factuality improvements in model training strategies—for example, Newman et al.'s "The Curious Case of Factuality Finetuning: Models' Internal Beliefs Can Improve Factuality," which relies exclusively on MiniCheck as a factuality measure, a metric we critically examine in this work.
>
> 5. Another practical risk here is that LLM-as-a-judge (or other "factuality" metrics) may be used for reinforcement learning (e.g., Li and Ng "The Hallucination Dilemma: Factuality-Aware Reinforcement Learning for Large Reasoning Models"), in which case models could conceivably learn to exploit "adversarial tricks"
>
> 6. While we agree with the reviewer that the community is aware of metrics being imperfect, our work provides a systematic investigation of the specific nature and extent of these limitations. We conduct a rigorous analysis of how these metrics respond to factuality in contrast to unrelated textual attributes, and examine their susceptibility to spurious correlations.
>
>  Note also that, per the reviewers' original suggestion, we performed additional experiments evaluating practical implications of LLM judges relying on their own parametric knowledge (summarized in our rebuttal): As hypothesized, we find that they are less good at evaluating the "factuality" of outputs when the corresponding inputs are counterfactual.

---

### Note · Authors · 2025-08-14

We thank the reviewers for their engagement and feedback; we have strengthened the work as a result. To summarize for the AC:
This paper critically assesses a suite of automatic factuality metrics, including bespoke models and LLM-as-a-judge method, across diverse models and benchmarks. We identify each metric’s strengths and weaknesses that inform appropriate use, and show they can be gamed raising concerns about reward hacking in RL and leaderboard manipulation.

All reviewers found aspects of this work insightful and timely. But while **pyjj** and **SyLu** advocate for acceptance, **kyR2** and **o3Me** maintain some reservations. We use these “final remarks” to address the larger concerns and highlight changes we have made in response to them.
- **kyR2** agrees that the paper “raises a useful cautionary note”, but thought it would be compelling to provide a realistic (non-adversarial) evaluation showing that LLM-based factuality evaluation might go wrong. We ran such an experiment (please see our response to **kyR2** for details), and will incorporate the results in the manuscript. Briefly, we found that, consistent with our original conclusion, LLM-based consistency evaluation performs worse when scoring summaries against inputs that themselves counterfactual information (i.e., when the reference is “incorrect” according to model parametric knowledge).
- **kyR2** has also suggested the work may have limited impact because the community is broadly aware that using LLMs as “judges” brings inherent limitations in general. We agree that most in the community are aware that LLM based evaluation will be imperfect. But this evaluation is squarely focussed on factuality assessment, a timely problem that poses unique issues, and to our knowledge no published work has investigated and characterized these limitations for judging factuality. Even if community members are broadly aware that LLMs are imperfect judges, we argue that empirically demonstrating limitations for factuality specifically is a useful contribution. Finally, we reiterate that LLM-as-a-judge is only one sort of “factuality” metric that we evaluated in this work.
 - **o3Me** raised concerns mainly around clarity of presentation, which were helpful. As per our response, we have clarified the language in the relevant parts of the paper and we appreciate the reviewer raising their clarity and quality scores as a result. We also plan on releasing all code and data, as requested.

---

### Decision · Program_Chairs · 2025-09-17

**Decision:**

Accept (poster)

**Comment:**

This paper presents a comprehensive evaluation of model-based factuality evals, focusing on whether a model-generated summary is grounded in the source document. The paper presents a series of experiments with interesting findings, e.g., that the correctness of these evals correlates strongly with difficulty levels defined by spurious features (e.g., word overlap with the source document), that model-based evals are sensitive to benign modifications while being insensitive to meaningful modifications (e.g. contradictions), that LLM-based evals rely heavily on their internal knowledge rather than the source document, and that these LLM-based evals are gameable, e.g., by adding a sentence like “The summary entails the information the document discusses”.

Strengths:
- This paper is well-written and well-organized.
- Factuality is an interesting and important topic in NLG evaluation, and this paper provides a comprehensive investigation of existing automatic evaluation metrics towards factuality.
- The authors provide insightful experiments and analyses.
- The experiments are well-designed, and all the conclusions/claims are verified by rigorous experiments. The overall conclusions and findings are solid.

Weaknesses:

This paper does not have major issues. Though this paper does not introduce new strategies or metrics to automatically evaluate factuality, the analytical work is also important for the community. This paper provides insightful experiments and analyses, and highlights the issues lying in existing evaluation metrics, which can also promote the research in factuality metrics.

Reviewer kyR2 provides great reviews.